# When in Doubt … Career Indecision, Mental Wellbeing, and Consultation-Seeking Behaviour—A Qualitative Interview Study among Students and Counsellors

**DOI:** 10.3390/ijerph182312604

**Published:** 2021-11-30

**Authors:** Katherina Heinrichs, Victoria Hermülheim, Laura Pilz González, Adrian Loerbroks

**Affiliations:** 1Institute of Health and Nursing Science, Charité—Universitätsmedizin Berlin, Corporate Member of Freie Universität Berlin and Humboldt-Universität zu Berlin, 13353 Berlin, Germany; laura.pilz-gonzalez@charite.de; 2Institute of Occupational, Social, and Environmental Medicine, Centre for Health and Society, Faculty of Medicine, University of Düsseldorf, 40225 Düsseldorf, Germany; victoria.hermuelheim@hhu.de (V.H.); adrian.loerbroks@hhu.de (A.L.)

**Keywords:** university students, doubts, career indecision, mental health, counsel, qualitative research

## Abstract

University dropout is often preceded by a phase of doubt whether to continue studying, either in general or just the given subject. Mental health problems might be interrelated with this phase of doubt. Counselling services at German universities could provide help, but do not seem to reach students in need. To explore the phase of doubt and possible (inter-)relationships with mental wellbeing among university students in Germany as well as their consultation-seeking behaviour, a qualitative interview study was conducted (2017–2018). Participants were students casting doubts on their studies (*n* = 14) and counsellors (*n* = 16) working with this target group. Examples of reasons for doubts were insufficient information, unfulfilled expectations concerning the subject, subjectively poor study conditions, performance problems, and lacking future perspectives. Mental health problems were subjectively intertwined with doubts, considered as both cause and effect. Counselling services were evaluated as hardly helpful by students and as being in need of improvement by counsellors. Suggestions as how to improve such services comprise a more specific and proactive way to approach students. By considering the phase of doubt before dropout, German universities can improve their support services to be more responsive to students and, thus, prevent dropout and mental health problems.

## 1. Introduction

University dropout is commonly defined as quitting the higher education system without a (first) graduation or subsequent re-entry [1] and represents a common issue in many countries, with considerable societal and economic implications [2]. International comparisons are hard to make because definitions and analysis procedures differ between countries [1,3]. The dropout rate for all types of higher education institutions and disciplines in Germany equalled 28% in 2018 [4,5]. Nearly one-third of student dropouts leave university because of performance issues (20%) or failed exams (11%) [6]. The dropout rate has steadily increased over the last years and is highest for bachelor students, who also quit earlier than students in other courses of studies. A total of 15% of university dropouts in Germany remain unemployed after six months [7], but it needs to be mentioned that dropouts often wait to exmatriculate until they find a job, which leaves unemployment undetected [8]. This development seems even more relevant given the severe lack of academically qualified professionals in a wide range of sectors [9,10].

University dropout is often preceded by a phase of doubt, a time of insecurity and questioning one’s career decision [11]. This phenomenon may not always lead to dropout, and affected students could decide to continue their studies or to change their subject [11].

Research on this topic considered various factors, such as dissatisfaction with study conditions, performance problems, or physical and psychological issues, which may cause university students to doubt their academic commitment and eventually drop out [11,12]. Though much attention has been paid to matters regarding the extent and causes of dropout, little attention has been given to the process prior to dropping out and, thus, to the initial uncertainties about the academic career path and the emotional as well as psychological burden this process of indecision entails.

Stress, psychological instability, and financial uncertainty are well-known issues among university students and evidently affect their wellbeing and academic performance [13,14,15]. In Germany, about every sixth student (17%) reported to have at least one psychological condition [16]. Students who cast doubts on their studies may face an even greater emotional and psychological burden that has not even been investigated yet. Their burden may be rooted in possible feelings of failure, loneliness, or incomprehension not only due to performance problems or failed exams, but also the disappointment in themselves [5,14,17].

Achievement emotions are the key element of the Control-Value Theory by Pekrun and Perry [17]. These emotions can be connected to studying or experiencing success or failure in exams, and they strongly impact students’ performance and wellbeing [17]. The (achievement) emotions, their causes, and their consequences are all elements of a cyclic process: for instance, failure may result in emotions such as anger, frustration, anxiety, hopelessness, shame, sadness, or disappointment. These emotions can have a detrimental effect on learning and performance, eventually leading to new failure—a vicious circle [17]. Causes for achievement emotions as well as their consequences can lie within a person or in their environment, e.g., in the form of task demands [17]. Therefore, individual (e.g., counselling) as well as situational interventions (e.g., adaptation of study conditions) could influence this cycle and its elements [17].

A variety of counselling services—ranging from general study support and subject-specific course guidance to psychosocial counsel—is offered to students at German universities. This is also intended to support students casting doubts on their studies [18]. However, counselling services are not used as often as expected and potentially needed in Germany [19]. This is not due to a lack of services or their inadequacy within and outside higher education institutions, but to the lack of awareness of their availability, barriers in accessing them, and a failure to identify the right time to contact professional counsellors [18]. However, the availability of services alone does not guarantee their use, especially when psychological ill-health can be hypothesised to be interrelated to the phase of doubts, as mentioned above. Therefore, options to reduce barriers of the utilisation of counselling services should be investigated.

Taking all of this into account, this paper aims at exploring the possible (inter-)relationships between the phase of doubts and mental wellbeing among university students in Germany as well as their consultation-seeking behaviour.

## 2. Materials and Methods

Given the explorative nature of the study, a qualitative study design was chosen. Semi-structured interviews with students and experts (i.e., counsellors) were conducted to gain insights from several viewpoints in terms of data triangulation.

### 2.1. Sampling

Students could participate in the study if they were enrolled at a university/university for applied sciences and (a) in doubt about their subject choice or their academic education in general or (b) had changed their subject at least once or (c) dropped out. An exclusion criterion was an exmatriculation due to failed exams.

Information on the study was distributed via social media, flyers, and posters at different institutions in the city of Düsseldorf, personal recruitment at information events for students casting doubts on their studies at the University of Düsseldorf, contacts with counselling services at different institutions in North Rhine-Westphalia, and personal contacts with fellow students of co-author V.H. Although the focus lied on the University of Düsseldorf, an effort was made to expand the target groups all over the region of North Rhine-Westphalia.

In addition, counsellors were asked directly to take part in the study. An inclusion criterion was that they worked with the abovementioned target groups. To explore a wide range of perspectives, counsellors of different institutions, such as universities, universities for applied sciences, and employment agencies, were approached. Furthermore, different counselling services, such as general and psychological counselling, specialised counselling for students in doubt, subject-specific course guidance, and services provided by employment agencies, were considered.

This recruitment strategy resulted in a non-probabilistic sample for both target groups.

### 2.2. Data Collection

Semi-structured interview guides were developed for the respective target groups, leading to different versions for students and counsellors. The interview guides were developed by V.H., discussed in the team (V.H., K.H., and A.L.), and pre-tested by V.H. No changes were made after pre-test.

As an “icebreaker question”, participating students were asked to explain why they chose their study subject. Afterwards, the following topics were addressed, each with an opening question and possible follow-up questions: (a) doubts concerning subject or academic career; (b) mental wellbeing; (c) utilisation of support, its form, and its subjective use; and (d) ideas for optimisation. All questions were asked in an open manner to encourage the participants to share their thoughts and stories.

The interviews with participating counsellors started with a question concerning their work in general. Afterwards, the following topics were addressed in a comparable manner as stated above: (a) students’ queries; (b) students’ mental wellbeing; (c) topics of consultation and support offers; and (d) ideas for optimisation.

At the end of the interviews, the participants were offered the opportunity to add further information. Additionally, sociodemographic characteristics were collected by a short standardised questionnaire, including gender, age, and questions concerning stress and psychological health (students) or counselling offers and professional careers (counsellors).

Data were collected in two phases: from April 2017 to February 2018 (students), and from March to June 2018 (counsellors). According to the participants’ preference, the interviews were conducted face-to-face or via telephone. V.H. conducted all interviews with constructive feedback by K.H. after two interviews. Data collection went on until thematic saturation was achieved, i.e., no new contents came up in the interviews. All interviews were audio recorded with the same device, which had no connection to the internet.

### 2.3. Data Analysis

Digital audio recordings of the interviews were professionally transcribed according to pre-set rules. The transcripts were analysed based on qualitative content analysis [20] using the software MAXQDA (2018) (VERBI GmbH, Berlin, Germany). Two different sets of categories were developed for the students and the counsellors. The main categories were deductively defined according to the main topics of the interview guide, e.g., “causes for doubts”. Subcategories resulted from inductive coding based on the interview material, e.g., “individual causes” or “study conditions”.

Four interviews (two of each target group) were analysed independently by V.H. and K.H. The resulting category systems were discussed and modified until a consensus was achieved. After the first round of analysis by V.H., the category systems and text samples were reviewed by K.H. and, after three rounds, additionally by A.L. The members of the study team had different professional backgrounds (i.e., medicine, psychology, public health, and epidemiology). Whereas V.H. was enrolled as a student at the time of data collection, A.L. was active in teaching. The engagement of three multidisciplinary analysists was meant to ensure intersubjective transparency, replicability, and discriminatory power of the categories [20].

## 3. Results

### 3.1. Sample Description

In total, 14 students from five different universities and 16 counsellors from seven different institutions participated in this study. Table 1 summarises the main characteristics of the sample. On average, the students were 24.4 years old, and the interviews with them took 28.5 min. Students represented a broad range of different study subjects and programmes. As a consequence of their doubts, one student had dropped out of university, and the remaining 13 continued their studies, either in their original subject (*n* = 6) or after a change of subject (*n* = 7). Out of the seven students who had changed their subject, two had discontinued their studies at first and completed an apprenticeship before returning to university.

The counsellors were between 31 and 65 years of age (mean: 42.1 years), and the interviews with them took 36 min on average. They worked in general counselling (*n* = 2), subject-specific course guidance (*n* = 3), psychological counselling (*n* = 3), or combined services (*n* = 5), amongst others. Five of the counsellors reported to be experts in dealing with students casting doubts on their studies and offered specialised consulting hours for this target group.

### 3.2. Why Students Cast Doubts on Their Studies

The participating students reported several causes for their doubts about their studies. At the beginning of their academic training, they mentioned receiving insufficient information beforehand on the targeted subject and future professional perspectives. As a consequence, they stated that they lost their interest in the study subject or that their expectations were not fulfilled. The issue of insufficient information and/or unfulfilled expectations typically emerged in the early semesters, as both students and counsellors explained alike:
*“Those who come to us in the first, sometimes also in the second or third semester, often held different expectations. A typical sentence then is: “I started with English studies because I was good at English, but it’s totally different here”, right? Or: “I studied psychology because I wanted to do something with people. Now I’m doing statistics all day long.” It’s the unfitting of expectations with academic reality. That’s those in the early phase.”*Counsellor, ID 1467

Some students also may not have had a concrete idea of what it meant to study on a university level, which seemed to be a greater problem for students from non-academic households:
*“Maybe it is because my parents didn’t study. They couldn’t tell me what to expect during my studies. So I did it on my own. And that didn’t turn out as expected, so to say.”*Student, ID 2574

According to some counsellors, perceived pressure from families and/or society was another issue for some students. Additionally, performance problems could lead to doubts, especially when people were good at school and faced first failures at university.
*“Performance problems in the early phase. “I thought I was good at maths, but I came to realise that is not enough.” That’s the early phase. Then there are phases in the middle of the studies when performance problems occur. “Yeah, I did well at the beginning, but now it’s becoming a bit trickier. Now I’m facing my third retake.””*Counsellor, ID 1467

Some students referred to their study conditions as factors contributing to their doubts. These include performance and time pressure, heavy workload, changing rules during their course of study, and a suboptimal atmosphere for teaching and learning. Some of these aspects resulted in performance problems and, thus, directly in dissatisfaction. As such, doubts often occurred during examination phases.
*“During the bachelor programme, I had a lot of doubts because I had my third try in mathematics. You can’t help to doubt then. Because you have to think about whether you want to continue to study, if you want to continue the training to be teacher or change the subject or head in another direction completely.”*Student, ID 3956

Comparisons with other students and their performance were mentioned to reinforce this dynamic. This effect seemed to be associated with the school type from which the students had graduated before entering university. In Germany, one can acquire the admission to study at a university level through different school institutions.

Besides the problems concerning personal interest and performance, future perspectives were crucial for some students to continue their studies. They reported their motivation decreasing when they did not identify with the targeted profession anymore. Furthermore, the perspective of an apprenticeship and a good salary after its completion was perceived to be appealing to some students and reinforced their doubts. Counsellors confirmed that students mainly began to doubt because of lacking perspectives towards the end of their studies. However, most of them continued to study their subject.
*“Mainly, it was about the daily work routine later on. I had the idea that a dentist’s life would be quite boring because you always do the same. And I just couldn’t imagine to do that for the rest of my life. Even if it is well-paid.”*Student, ID 1546

Health problems were only mentioned—but not specified—by one student as a cause for doubts. Further demanding factors which were not consistently perceived to be linked to doubts were financial issues, dependence on parents, lacking social contact, living conditions, and commuting to university as well as society’s wrong perception of student life, which is considered to be fun.

### 3.3. Students’ Mental Wellbeing in the Phase of Doubts

Most of the students reported their mental health to be impaired during the time they had doubts about their studies. Sadness seemed to be related to non-fulfilled expectations and performance problems.
*“Sad in any case because it didn’t work out as I had imagined, so that somehow I often thought: “It works for all the others. Why doesn’t it work out for me?””*Student, ID 7634

Doubts concerning future perspectives were often associated with anxiety and feelings of insecurity, e.g., about the future in general, financial security, and the fear of not finding a suitable profession later. Especially when students experienced performance problems, they mentioned feeling anxious about the prospect of not being able to engage or pursue their chosen profession.
*“Mainly the feeling that I don’t want it to be taken away from me. I have a place in such a sought-after study programme. It is exactly what I want to do professionally afterwards. Yeah, I felt existential fear. Mainly because I had no alternative. I wouldn’t have really known what to do if I had really been exmatriculated.”*Student, ID 3854

Furthermore, students feared failing when facing their third (and, at German universities, final) attempt to pass an exam. In some cases, they reported to have lost their self-confidence and doubted their abilities.
*“I think, I didn’t have any self-confidence then because I didn’t believe that I could pass anything. At that time, I had an additional mathematics exam. And I immediately thought: “I didn’t pass, and that’s it.” Although I did pass with very good marks. But somehow, you didn’t believe in yourself anymore. I still feel this way a bit. I think that once you’ve been through that—“You aren’t good enough!”—, you think everything is bad, and you begin things with a bad feeling instead of saying: “Yes, that was good.””*Student, ID 3956

Additionally, some students described themselves as irritable, helpless, and angry, either with themselves or with the study conditions that they considered to be beyond their control. Apart from psychological consequences, some students experienced changes in their private life, e.g., they avoided social contact, or when they did engage in leisure time activities, they had a bad conscience.

Concerning their studies, some students described that they needed more time or lacked motivation for learning. Furthermore, some tended to avoid to go to their lectures and seminars or even to their exams, which slowed down their whole educational career.
*“Well, I think, the motivation wasn’t high in general. You stayed away from the lectures because you just didn’t feel like it. Or maybe that didn’t encourage you, but you were simply more desperate.”*Student, ID 5923

Moreover, some students reported to have a bad conscience because of their problems at university and feared disappointing their parents. Some felt great shame and avoided to talk about their studies altogether, sometimes because they did not want to be influenced during their decision-making process. Recalling this phase, some students stated having repressed their doubts and the related problems.

From the counsellors’ perspective, health problems are a key element to doubts, including chronic conditions, disability, and psychological illnesses, mainly depression. Counsellors described an interrelationship between doubts and psychological problems: they explained that they perceive doubts to potentially emerge from mental ill-health. High pressure and work overload were assumed to lead to disappointment, self-doubts, and feelings of failure. As a consequence, conditions like depression and different forms of anxiety might develop and reach a clinically relevant level that hinders the affected students from finding solutions to their problems. This phenomenon was mainly observed for students who were facing issues for the first time during their studies.
*“There are many students with performance problems who I would also say leave a depressed impression, a certain depressive mood–without suggesting that you can make a diagnosis after a ten or fifteen minutes counselling session. But many are depressed facing the upcoming failure of their studies (…) That’s obvious. And that their lives are shaped by it negatively. You cannot deny this impression.”*Counsellor, ID 7446

However, the participating counsellors stated that students who started doubting in the early semesters seemed psychologically stable. The same holds true for students who were advanced in their decision-making process.

### 3.4. Utilisation of Counselling Services

Out of the 14 students that were interviewed, only five had approached or were currently in contact with a counsellor at their university. For instance, they sought general or subject-specific counselling to get information on formalities, e.g., in connection with a change of subject, or their professional prospects. One student sought psychological counsel, another one engaged in a mentoring programme. The latter used the offer quite early and in a preventive sense, as he explained, to organise his educational and private activities and to learn how to prioritise and to use his resources.
*“I sought help in the mentoring programme. And then, we practically made a big scan: “What does your week look like? What’s your schedule like? On which days will you do which tasks?” You have to look at the whole life situation. (…) With the mentoring, I saw the big picture. (…) And I think, that this is a completely different way to approach the matter.”*Student, ID 2954

The two students who took part in ongoing counsel experienced them as quite positive. However, others who approached general or subject-specific counsel offers did not subjectively benefit from the meetings. They felt like their problems were not detected properly and that they did not get the help they hoped for, e.g., because they found the sought-after information themselves before they went to the counsellor. Counsellors stated that some students got in touch for support rather late, especially when they suffered from psychological conditions. Possible reasons for this were reported to be fear of stigmatisation or the non-binding nature of the counsel service.

When students in doubt reached out for professional counselling, the counsellors faced different sets of expectations. Some students just explained their problem, but could not state a certain query. Other students sought for information on professional alternatives or wanted their own decision to be confirmed. The psychological counsellors described that students in doubt wanted to analyse the problem of their situation and sought support from a neutral person with the goal to improve their general situation and mental wellbeing. It was also highlighted that some students expressed unrealistic expectations, such as asking for a quick and simple solution to their problem or for their counsellor to make the decision for them.

Regarding the benefit for students, the counsellors’ view was that students may learn self-efficacy, self-reflection, and problem-solving through several counselling sessions. However, some counsellors stated that the success of such a coaching depended on the student’s mental state.
*“You can achieve this very quickly with psychologically healthy people, partially in case of moderate problems, but you can hardly achieve this among those with a psychological condition because they are so deeply stuck in their negative spirals. How much their problem-solving competence develops strongly depends on them.”*Counsellor, ID 1467

According to the counsellors, mechanisms to avoid emotional and work overload and, thus, failures and, additionally, to accept support and to ask for help in the future could also be conveyed through their counselling services. Furthermore, counselling itself offers a safe space to reduce feelings of shame where students can learn that they are not alone with their problems. Moreover, they get information and an overview of possible support offers. However, no formal evaluation of the counselling services was reported.

One of the main reasons mentioned why so few students used the services provided by their universities was that they mostly sought help in their private social environment. Family members, especially their parents, friends, and partners offered practical help, emotional support, and advice in the decision-making process. Nevertheless, when it came to specific problems of the training at university, the participating students stated that they preferably approached fellow students as they felt understood by them.

When asked for specific reasons why they did not try to get professional counselling, the students explained that they only had vague knowledge of the counselling offers at their universities, e.g., regarding the topics that could be addressed, and what function the counsellors had. They also mentioned wanting to get help by someone close to them, questioning the benefit of professional counselling, or feeling uncomfortable to discuss personal problems with outside parties. Additionally, they felt that their problems were not acute and, thus, perceived their need for professional counselling not to be strong enough.

Despite the fact that counsellors reported the specific offers for students casting doubts on their studies to be well-attended, they criticised that their offer might not have been perceived as helpful by all students because of a confusing abundance of information. They reported affected students having problems finding out about their specialised offer. Furthermore, some counsellors criticised the specialisation of counselling services and advocated integrated counselling because individuals may use the open consultation hours for students casting doubts on their studies although their mental health state would require a psychological consultation.

Though some counsellors had the opinion that specialised offers were not necessary, others highlighted the importance of individual counselling for students casting doubts on their studies because of their being a demanding target group and counselling them was time-consuming. Ongoing individual counselling prevented a prolongation of the decision-making process and led to well-considered decisions.

### 3.5. Ideas for Optimisation

The participating students wished to get more information on the counselling services at their universities, preferably at the beginning of their studies and in a proactive manner. They mentioned that universities should identify students with problems and invite them to meetings. Counsellors should provide information on how to plan the course of the studies, e.g., in a mentoring programme. However, most of these suggestions are already implemented at the respective universities, which stresses the necessity for more awareness of the existing counselling services.

The counsellors described students casting doubts on their studies as a heterogeneous group and, thus, not easy to address. They advocated specific information events to raise awareness for the problem. The existing offers need to be promoted more offensively according to counsellors. In accordance with the participating students, the counsellors demanded a more proactive strategy to address affected students. One of the participating universities followed such an approach already, and it became clear that it targeted students who would not have been reached before, e.g., because they were no longer present on campus and, thus, could not find information material by chance.

Some of the participating counsellors thought that their networking with other counselling services and/or staff should be optimised to ensure professional exchange and following-up the students’ development. This was already done at some institutions in the form of cooperation projects with internal and external partners. However, this collaboration was reported to be a new approach, so feedback on the implementation could not be given. Nevertheless, some counsellors mentioned receiving little information on students they transferred to colleagues at other institutions.

## 4. Discussion

This study aimed at exploring the possible (inter-)relationships between the phase of doubts and mental wellbeing among university students in Germany. It also looked into their consultation-seeking behaviour. The key results are summarised in Figure 1.

Students and counsellors alluded to numerous perceived reasons for students to cast doubts on their studies. In general, reasons for doubts range from insufficient information and unfulfilled expectations concerning one’s subject, subjectively poor study conditions, and performance problems to lacking future perspectives. This can lead to feelings of sadness, a lack of self-confidence, anxiety, and to doubts whether either to continue studying or to change the subject or to completely drop out. Interrelationships with mental and social wellbeing were found, e.g., in the form of psychological illness or behaviours like social withdrawal.

The subjective causes for doubts found in our study align with those for university dropout in the literature, especially regarding performance issues and unfulfilled expectations [6]. Similarly, our results are also in line with prior research stating that stress, mental wellbeing, and academic performance are closely intertwined [13,14,15]. Findings that showed associations of feelings of failure and self-disappointment with performance problems or failed exams [5,14,17] were also underlined by this research. The Control-Value Theory provides a framework to understand the interrelationships between emotions and their causes and consequences, including doubts. In case of failure, negative achievement emotions affect learning and performance reciprocally leading to new experiences of failure and, in our case, doubts [17]. Our results contribute to the research field by moving the focus to the doubting phase prior to dropout. The causes and effects of such a doubting phase are specified in Figure 1, which highlights the interrelationships between the causes of casting doubts, the respective emotional reactions, and the effects on mental and social wellbeing of having such doubts.

Although doubts exert negative effects, only few students used the counselling services offered by their institutions. This under-utilisation is also in keeping with prior research reporting that the counselling services provided are not used as much as expected or potentially necessary [19]. Some students participating in our study reported that they felt ashamed because of their doubts and, therefore, stopped talking about their studies in general. Others stated that their social network was an important source of support and advice. Counsellors assumed that some students did not use counselling in the long term because of its non-binding nature.

When students in our study sought professional counsel, they considered the services offered to be hardly helpful. Nevertheless, counsellors emphasised that their services, especially coaching that takes several sessions, could have positive effects. Anastasio et al. [18] also highlighted the benefits of counselling services and described the problem not as one of availability—because the services do exist—but of poor accessibility and actual usage. In our study, the actual utilisation of counselling services was affected not only by problems of awareness, but also by dysfunctional expectations towards such services. Furthermore, our results confirm the cyclic process of causes, achievement emotions, and their consequences, including doubts, proposed in the Control-Value Theory [17]. What sounds like a vicious circle has an advantage: interventions can address any of the components of the circle—with positive effects on all of the following elements [17]. Causes for emotions can lie within a person as well as outside, e.g., in the form of task demands and environments [17]. This allows the conclusion that not only individual interventions, such as counselling, could influence the cycle, but also environmental interventions at universities, e.g., an improvement of study conditions to prevent students from being trapped in such a dysfunctional cycle.

An example for prevention on the organisational level could be a more proactive and automatised way to approach students at risk. For instance, the time of consultation-seeking was shown to be associated with the success of the counselling sessions. It was clearly mentioned that students should approach counsellors earlier when they start experiencing mental health problems or begin having doubts about their studies. To tackle the delayed and assumed under-utilisation of services, it was encouraged by students and counsellors alike that counselling services should use a more problem-specific and proactive way to approach the affected students. However, the counsellors’ opinion on the best way to approach affected students was not uniform: though some were in favour of specialised services, such as open consultation hours for students in doubt about their studies, others criticised this approach and advocated more integrated general counselling. Although changes at the service level are essential, a more proactive and responsive approach cannot solely rely on changes in counselling. Students must also be better informed in order to be able to identify and recognise possible problems and, thus, seek help at an earlier stage. As a consequence, it would be advisable to focus on vulnerable groups at the beginning of their studies, such as students from non-academic households or students who received their admission to study at university from certain schools types, such as the German “Gesamtschule” (integrated comprehensive school) [1]. In addition, our results showed that different reasons for doubts emerge at different stages of the academic education and, thus, the nature of counselling services should remain flexible. Counsellors should, therefore, approach students with their specific problems in mind—for instance, offer information, advice on study conditions, and career guidance in early semesters and information on future perspectives in later semesters. Furthermore, it is essential that counselling services also address students who are determined to drop out. Here again, services should adapt to the specific needs of those affected. Counsellors should provide information on options outside the academic setting, such as opportunities to find a job directly or get an apprenticeship where the already acquired skills can be used. This is of special importance considering the high unemployment rate among former students and the lack of qualified professionals in a wide range of sectors [7,8,9,10].

Our data collection took place in 2017 and 2018, that is, before the COVID-19 pandemic. Given the current global developments in the years 2020 and 2021, the already high incidence of student dropout is expected to further increase not only in Germany, but also worldwide [21]. Therefore, actions need to be taken fast, and counselling services need to focus on newly emerging problems. Additional research is needed to identify problems students might have especially at the beginning of their studies, e.g., due to the effects of month-to-year-long online teaching at schools and universities, and to find appropriate solutions for them. Furthermore, quantitative research is needed to statistically confirm the relationships we found and, thus, to generate evidence that is generalizable in statistical terms (e.g., the prevalence of specific views or challenges). Longitudinal studies on the effects of emotions, their causes, and academic achievement could help generate suiting and impactful interventions, in particular, concerning the phase of doubts.

### 4.1. Strengths of This Study

Our study provides first insights into a new research field by addressing the phase of doubt and indecision prior to university dropout in connection with mental wellbeing. By following an approach of data triangulation, the perspectives of students casting doubts on their studies and counsellors working with them could be taken into account to pursue our study aim. Analysis was performed in multiple rounds and by a multi-professional and experienced team of researchers (i.e., in terms of qualitative research with a focus on mental health among students [22,23,24]) to ensure intersubjective transparency.

### 4.2. Limitations of This Study

First, data collection took place in two different phases, and a convenience sample was drawn through different ways of recruitment. Selection effects cannot be ruled out, e.g., students who dropped out of university were difficult to approach. To face this issue, some effort has been made to recruit participants via employment agencies. Personal acquaintance with the interviewer (V.H.), who was enrolled as a student at the time of data collection, could have led to biased information. On the one hand, it is conceivable that the respective participants withheld certain personal information from the interviewer. On the other hand, it is possible that personal acquaintance helped both sides to be more open in the interview situation. The sample size of *n* = 30 might appear small, but is average for qualitative research among university students [25,26,27,28]. Due to the qualitative approach, our results are not generalizable in a statistical way, but they deliver valuable subjective insight from different perspectives.

Second, the interviews were conducted several years ago (in 2017 and 2018). The situation of the students at German universities may have changed mainly due to the COVID-19 pandemic, i.e., through the introduction of digital teaching approaches and increased feelings of loneliness [29,30]. On an international level, online learning was shown to be associated with decreased satisfaction [31,32], performance problems, and mental health issues [33] as well as perceived stress [34]. It is likely that further performance and mental health issues became more pronounced in addition to issues of lacking information, subjectively poor study conditions, or future perspectives. Furthermore, social distancing and online teaching might have detrimental effects on students’ social wellbeing, such as increased feelings of perceived stress, loneliness, and isolation, evidently affecting their mental health and, thus, their academic performance [29,30]. Additional research in the COVID-19 era is thus warranted.

Third, the interviewer (V.H.) initially lacked experience in conducting interviews, which might have been detrimental for the data quality and information richness of the interviews. To deal with this issue, V.H. was closely supervised by K.H. and A.L., who are both experienced in conducting and analysing qualitative interviews [22,23,24,35,36].

Fourth, most of the interviewed students shared their experience retrospectively. Recall bias might have influenced the given information. It is conceivable that relationships between causes and effects were seen more clearly from this perspective. Moreover, the interview partners might have understated the potential negative psychological impact of their doubts. To account for this issue, the interview guide was constructed with care to mentally bring the students back into their phase of doubt and insecurity.

Fifth, no data on clinically relevant diagnoses were obtained. Information on psychological conditions were only given by students and counsellors. Therefore, conclusions concerning clinically relevant psychological illness must be drawn with caution.

Fifth, participants were not involved in the phase of data analysis. Misunderstandings cannot be ruled out, but their probability was reduced by the involvement of three researchers in data analysis.

## 5. Conclusions

With this study, we aimed at gaining insight into how universities can develop support systems that can deal with doubting students more effectively and ultimately prevent them from dropping out. To achieve this aim, we explored the phase of doubt that precedes university dropout through the lens of not only students casting doubts on their studies, but also of counsellors who work with them. Theoretical models, such as the Control-Value Theory, recognise the cyclical relationships between emotions, their causes, and their consequences for academic performance and achievement. Although possible causes of emotions are found at the individual level, they also influence and are influenced by their environment. Therefore, in order to affect this cycle, interventions should take place at both the individual and environmental level. Regarding the latter one, German universities could provide more proactive, responsive, and timely support services that focus on students’ experiences and needs and, thus, intervene before doubts evolve into university dropout and mental health problems. Especially now—in times of online learning and social distancing—there is an urgency to focus on students’ issues, doubts, and mental health problems, which might have been growing under the influence of the COVID-19 pandemic.

## Figures and Tables

**Figure 1 ijerph-18-12604-f001:**
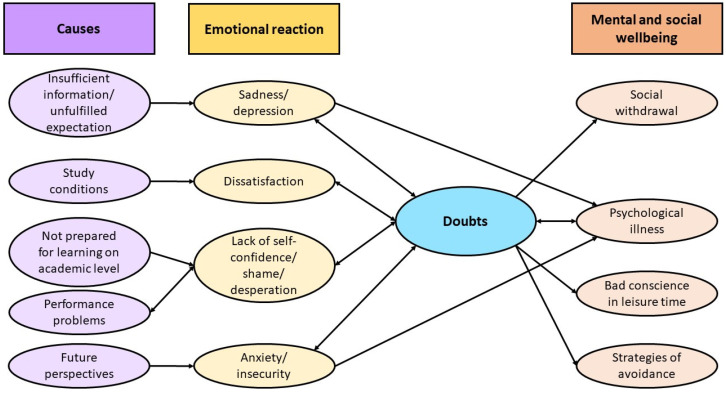
Causes for doubts, emotional reactions to them, and their impact on mental and social wellbeing; one arrowhead: perceived cause and effect, two arrowheads: perceived reciprocal influence.

**Table 1 ijerph-18-12604-t001:** Sample characteristics.

Variables	Mean (Min–Max) or *n*
**Students (*n* = 14)**	
Interview duration in minutes	28.5 (15–81)
Codes linked to interviews	670
Age in years	24.4 (22–28)
Semesters at university	9.2 (2–12)
Gender	
Female	7
Male	7
Subject	
Medicine	3
Teaching	3
STEM	3
Liberal arts	4
Other	1
Programme	
Bachelor	7
Master	4
State examination	3
University type	
University	13
University of applied sciences	1
Study course	
Change of subject	7
Dropout	1
Continuation	6
**Counsellors (*n* = 16)**	
Interview duration in minutes	36.1 (20–50)
Codes linked to interviews	787
Age in years	42.1 (31–65)
Gender	
Female	13
Male	3
Professional background	
Psychology	5
(Social) pedagogy	4
Other	7
Employer	
University	11
University of applied sciences	3
Employment agency	2

SD = standard deviation; STEM = science, technology, engineering, and mathematics.

## Data Availability

The data presented in this study are available on request from the corresponding author. The data are not publicly available due to their sensitive nature.

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
