# Peer review of "When in Doubt … Career Indecision, Mental Wellbeing, and Consultation-Seeking Behaviour—A Qualitative Interview Study among Students and Counsellors"

_ijerph, 2021, doi:10.3390/ijerph182312604_

Round 1

Reviewer 1 Report

Topic is interesting, from guidance perspective and mental health of university students. In addition, circumstances surrounding the lack of counseling, doubts and the risk of dropping out the university is a subject that has been little investigated.  That is why it seems appropriate and useful in itself and it is necessary to increase the knowledge on this subject, for the students and to be able to know its relations with the social and economic context.

The title is original, but it seems that data are available for all university students in Germany. But this is not the case, because the study focuses on the University of Düsseldorf with 14 students and 16 counselors.

The abstract indicates the possible cause-effect relationship between mental health and doubts about academic decisions, but the paper is not developed in that way.

The introduction includes a lot of data, but there is no comprehensive and in-depth state-of-the-art analysis. It is true that dropout has been studied, but not the process developed by people before making the decision to drop out.

The sample size is very small, although this does not have to be a big problem in qualitative exploratory research if there are many units of analysis (sentences). But nothing is indicated on this in the paper.

Thus, trying to generalize with such a small sample of informants is risky. This section reports about sampling, not on sample. Information about the sample can be found in another section (sample description).

The information collection and data analysis section is correct. Regarding the procedure, only the process of elaboration of categories is indicated in a general way, but without specifying it.

The results are well presented, and comprehensively discussed in relation to doubts, mental health, and seeking counseling, although there are data and comments on them that are superficially treated.

The results are well presented, and comprehensively discussed in relation to doubts, mental health, and seeking counseling, although there are data and comments on them that are treated without going in depth.

The improvement section is good and it is good to improve counselors interventions and the student-counselor relations in order to increasing the consultations and the follow-up of the students.

The discussion section and the strengths and weaknesses section are also of interest, but the conclusions section is insufficient.

Author Response

Please see file "211101_rev_responses".

Reviewer 2 Report

The manuscript deals with university students' dropout and its counseling approach. The topic is not new, but its research approach and contribution by the authors can make a contribution to the journal. There are several minor issues for revisions.

  • Research background and motivation should be extended.
  • I suggest including a figure that shows research concepts and their focus.
  • The interview date is relatively old (2017-2018).
  • The number of subjects is too few.
  • More recent references are required for comparative analysis.
  • More quantitative research results are required (e.g., statistical analysis).
  • In conclusion, I suggest adding future research directions based on limitations.

Author Response

Please see file "211101_rev_responses".

Reviewer 3 Report

Thank you very much for the opportunity to review the manuscript entitled „When in doubt … Career indecision, mental wellbeing and consultation-seeking behaviour – a qualitative interview study among university students in Germany “. The article is overall well-written, and it is of interest for the readership of International Journal of Environmental Research and Public Health as it discusses the phase of doubt preceding university dropout.

I would like to encourage authors to consider one key issue to be improved. I believe that after addressing this issue, the paper will have a value for this journal. I hope that my comments are useful for authors, as they further develop the manuscript.

At this point, the article lacks a theoretical framework. While the authors stress on the explorative nature of their study, I feel that considering a theoretical reference could improve the value of their work. Thus, I would like to suggest to the authors to use the framework provided by the Control-Value Theory of Achievement Emotions. It offers an integrative approach for describing the interrelations between emotions, their antecedents and their outcomes. Some suggested resources are:

Pekrun, R., & Perry, R. P. (2014). Control-value theory of achievement emotions. In R. Pekrun & L. Linnenbrink-Garcia (Eds.), International handbook of emotions in education (pp. 120–141). Routledge/Taylor & Francis Group.

Pekrun, R. (2018). Control-value theory: A social-cognitive approach to achievement emotions. In G. A. D. Liem & D. M. McInerney (Eds.), Big theories revisited 2: A volume of research on sociocultural influences on motivation and learning (pp. 162–190). Information Age Publishing.

Author Response

Please see file "211101_rev_responses".

Round 2

Reviewer 3 Report

Taking into account the revisions and the answer provided by the authors to my comments, I believe that the manuscript has been improved and now warrants publication in Journal of Environmental Research and Public Health.